# Have Policies Tackled Gender Inequalities in Health? A Scoping Review

**DOI:** 10.3390/ijerph18010327

**Published:** 2021-01-05

**Authors:** Nuria Crespí-Lloréns, Ildefonso Hernández-Aguado, Elisa Chilet-Rosell

**Affiliations:** 1Preventive Medicine Service, Hospital General de Elche, 03203 Elche, Spain; 2Department of Public Health, Universidad Miguel Hernández de Elche, 03550 Alicante, Spain; ihernandez@umh.es (I.H.-A.); echilet@umh.es (E.C.-R.); 3CIBER Epidemiology and Public Health (CIBERESP), 28029 Madrid, Spain

**Keywords:** gender policies, health inequalities, gender mainstreaming, gender equality

## Abstract

Gender is recognized as one of the most relevant determinants of health inequalities. This scoping review sought to identify and analyse policies, either implemented or formulated as proposals, which aimed to reduce gender inequalities in health. We searched Medline, Web of Science, and Scielo. Of 2895 records, 91 full text articles were analysed, and 33 papers were included. Of these papers, 22 described the process of formulation, implementation, or evaluation of policies whose aim was to reduce gender inequalities in health; six focused on recommendations, and the remaining five dealt with both issues. Our review showed that the policies aimed at reducing gender inequalities in health, either implemented or formulated as proposals, are scarce. Moreover, despite some success, overall progress has been slow. The studies show failures in design and particularly in the implementation process. We found a lack of awareness and capacity in the policy-making progress, under-financing, bureaucratization, shortage of relevant data, and absence of women’s participation in decision-making. Therefore, an emphasis on the design and implementation of gender-sensitive policies seems essential to advance gender equality in health. This scoping review gathers evidence to support the design of such policies and recommendations that can facilitate their implementation.

## 1. Introduction

Gender is recognised as one of the most relevant structural determinants of health inequalities [1,2]. Gender determines health behaviours, care practices, health responses, and ultimately health outcomes. Men die younger than women. In some countries, that life expectancy gap is 11.7 years or more, and it is in part explained by their higher rates of consumption of tobacco and alcohol, their likelihood of death from violence, deaths from road injuries, and deaths by suicide. Even though women might live longer, they suffer longer with chronic diseases. This higher risk of morbidities is linked to their reproductive roles, a lower status in society, and gender norms that impair their agency and bargaining position in sexual relationships inequalities [2].

The World Health Organisation (WHO) and the United Nations itself have urged countries to introduce the gender perspective into their health systems and all public policies to combat health inequalities [3,4]. This statement entails the assessment of the implications for men and women of any planned action, including legislation, policies, or programmes, in all areas and at all levels [5].

Since the 1990s, international agreements and proposals for action at different levels of health systems, as well as in other policies that impact on health, have multiplied. International commitments to advance gender equality have brought about improvements in some areas: child marriage and female genital mutilation have declined in recent years, and women’s representation in the political arena and in managerial positions is higher than ever before [6]. In order to facilitate the implementation, numerous tools have been developed such as guides, manuals, and indicators [7,8,9,10] that constitute a wide range of proposals, from global health policies to actions directly aimed at health care services. However, despite the efforts and commitments, gender inequity in health persists today [11], and our health systems continue to be a fundamental source of these inequalities [12]. 

The incorporation of the gender perspective into health systems in a way that effectively has an effect on health and health equity requires at least two actions. First, analyse the available proposals that can plausibly have effective results, and second, identify those experiences that have already been implemented. There are some review studies on policies and experiences of gender sensitivity in the health sector, such as that of Celik et al. (2012) [13], which focused on the main barriers and facilitators in the implementation of gender mainstreaming in health systems. It revealed that isolated actions focussed on a single level and a single difficulty are ineffective in reducing these inequalities, and it is necessary to have broad approaches that act simultaneously on awareness, on the structural and organisational level, and on the acquisition of professional skills. The work of Payne (2012) also addressed through a discussion paper the obstacles to this process in health policies. She pointed out that gender-based power hierarchies could be at the root of the problem [14]. On the other hand, in 2008, Ravindran et al. carried out a review of the evidence on experiences with a gender perspective in the health sector in a comprehensive way [15]. This 13-year review also noted the wide “gap between intention and practice” in the process of gender mainstreaming in the health sector and made recommendations to facilitate it.

It is undoubtedly necessary to incorporate the gender perspective into health care because of the positive influence that the involvement of health professionals can have on society as a whole. However, previous research showed that evidence on the relevance of gender inequalities in health care have not gone along with research on effective interventions for the application of gender-oriented clinical interventions [16]. Nevertheless, there is a need for policies at the population level that incorporate the gender perspective into population-oriented health policies and into non-health policies that have an impact on the health of the population. In order to move forward, we need to assess the available experience in reducing gender-based inequities and, where possible, learn how to scale up successful experiences. Our objective here is to identify useful information that can be used to overcome or reduce gender inequalities in health. Existing knowledge regarding gender-sensitive policies must also be updated to provide as strong a foundation as possible on which to continue building gender equity in health. To this end, this work aims to identify and analyse the policies aimed at reducing gender inequalities in health, either implemented or formulated as proposals that exist in the literature at the international level.

## 2. Materials and Methods

This scoping review was developed following the Arksey and O’Malley’s methodological framework, which we used to guide our reporting where possible. A scoping review allows us to examine the nature, range, and extent of the available evidence on a specific topic in fields where evidence is emerging and based on studies with diverse designs. In addition, it was chosen because it is an appropriate type of study to summarise and disseminate the findings that guide the actions of political decision-makers (Arksey and O’Malley) [17]. Rather than being guided by a highly focused research question and search for a particular study design, the scoping study seeks to identify all relevant literature regardless of study design. In summary, the main stages of a scoping review are (1) to identify the research question; (2) to identify relevant studies; (3) study selection; (4) charting the data; and (5) collating, summarising, and reporting the results.

### 2.1. Search Strategy

A literature search in May 2019 was conducted in the main bibliographic database in social and health sciences (Medline through Pubmed, Web of Science, and Scielo). The search strategy was agreed upon and reviewed by the three researchers to ensure the highest sensitivity. The studies had to contain in titles/abstracts terms related to “politics”, with “health policy”, with “community health” or with “health system” and in turn contain the term “gender” in the title. We limited the search by type of document (article, review, clinical trial, or report) and by language (Spanish, French, English, and Portuguese). The full strategy is detailed in Appendix A. Subsequently, to increase sensitivity, we performed a handsearch in the reference list of the final sample of included articles. 

### 2.2. Study Selection

We included studies that deal with policies (health and also non-health policies) aimed at reducing gender inequalities in health. Studies must provide a description of the formulation, implementation, and/or evaluation of these policies, or formulate recommendations. Studies needed to allude in their introduction and justification to gender inequalities and the reduction of gender inequity in health. We included policies if they justify that they would have an impact on health. Regarding the studies that formulated recommendations, we only included those in which one of the objectives of the study itself was to produce them using an explicit methodology (such as a consensus of experts, a scientific review, or results of an analysis aimed to produce recommendations) and recommendations that appear in a specific section of an article that evaluated the implementation of a policy.

The exclusion criteria were as follows: (a) policies or recommendations at the individual level (direct relationship between professionals and individuals or groups of individuals) and did not proposed actions at a structural level; (b) isolated recommendations in discussion or conclusions and not included in the results section; (c) policies or recommendations to address gender equity among health system workers; (d) editorials, letters to the editor, perspectives, comments, books or book chapters, or general guidelines without a detailed methodology and procedure to elaborate them. 

All search results were first screened based on title and abstract by two reviewers, and a third reviewer resolved discrepancies. We performed an initial analysis of four papers in order to homogenise the full-text inclusion criteria. After that, the full text of potentially useful records was reviewed by two researchers, and in order to increase sensitivity, a third reviewer confirmed his/her agreement with excluded ones. In the next step, we checked the list of references of included articles in order to identify potential articles to analyse. A detailed flow diagram of study selection is shown in Figure 1. 

### 2.3. Data Extraction and Synthesis

Two reviewers carried out data extraction using a standardised data extraction form. First, we extracted information regarding the main descriptive characteristics of the articles (objective, methods, year, country, type of policies/recommendations, and health problem). In articles dealing with the evaluation of policies, we extracted information regarding the variables analysed, results, and conclusions of each of the studies. We performed a descriptive analysis of the information obtained from items previously described by identifying common aspects in the analysis of policies and recommendations.

## 3. Results

Having removed duplicates (606), we screened 2895 titles and abstracts retrieved through a database search. After applying inclusion criteria, 29 studies were included in our scoping review. In addition, after checking the reference lists of these papers, four additional references were included in our analysis (Figure 1). Therefore, 33 articles were finally included in our review [18,19,20,21,22,23,24,25,26,27,28,29,30,31,32,33,34,35,36,37,38,39,40,41,42,43,44,45,46,47,48,49,50]. These articles were published between 2002 and 2018. Eighty percent of the articles were published after 2009. Of the 33 papers, 22 described the process of formulation, implementation, or evaluation of policies whose aim was to reduce gender inequities in health; six focused on recommendations to reduce gender inequalities; and five papers covered both issues.

### 3.1. Policy Formulation, Implementation, and Evaluation

The 27 papers that addressed policy covered at least one of the following areas: policy formulation (11), policy implementation (16), or policy impact (13). Regarding geographical distribution of the studies, eight included policies in Europe, four included policies in Latin-America, three included policies in North America, and eight included policies in multiple countries around the world. There was also one study conducted in Oceania, two in Africa and one in Asia. 

#### 3.1.1. Policy Formulation

Of the eleven studies that analysed policy formulation, seven [18,25,29,34,37,38,41] addressed health policies and five [25,28,43,44,48] focused on non-health policies. Regarding methodology, eight papers used qualitative methods, two papers used quantitative methods, and one was a systematic review (Table 1).

These studies were mostly critical of the policy formulation process [18,25,29,37,38,41,43,44]. The criticisms focused on the failure to include the importance of gender inequalities in health in the justification of policy [38]. Likewise, the authors pointed out not only the oversimplification of the causes of inequalities [41,44] but also the lack of strategies for addressing them in policy planning [43]. Similarly, three papers deplored the fact that gender policies only referred to those related to maternal, child, and reproductive health [29,37,38].

Five other papers highlighted the absence of a women’s rights-based approach in the formulation of the policies [18,25,29,43,44] and suggested that their formulation could cause the strengthening of gender norms and roles [43]. Finally, one of the main problems in policy design is the lack of participation of civil society and expert women’s associations [29,41,44].

One of the studies highlighted the strong potential of the plan for gender equity integration of Sweden to empower women, particularly in the areas of unemployment, education and distribution of resources and agency [28].

#### 3.1.2. Policy Implementation

Sixteen studies analysed the processes of implementation of policies focused on gender inequities in health. Twelve addressed health policies [18,25,29,33,35,37,38,39,40,41,42,50] and seven addressed non-health policies [19,24,25,32,36,39,40]. Most of the papers (10) used qualitative methodology, two used quantitative methods, three applied both methods, and one was a systematic review (Table 2).

They identified different implementation problems. The main barriers were: (a) lack of political commitment [18,29,35,39,40,42,50] (b) deep-rooted gender norms and laws that leave women unprotected [18,25,32,39]; (c) insufficient awareness and capacity [19,25,33,39,40,41,42,50]; (d) changes in organisations that hinder implementation [29,40,41,42]; (d) insufficient funding and human resources for implementation [25,39,40,42,50]; (e) bureaucratisation [38,41,42]; (f) lack of gender indicators for evaluation [29,35,37,39,40]; (g) insufficient collaboration and absence of alliances [37,50]; and, (h) non-participation of civil society [29,39,40,41].

The studies by Burke et al. (2017) [24] and Keippel et al. (2017) [36] showed the benefits of generating alliances and collaborations. Burke et al. analysed the community coalition created to develop and incorporate strategies for addressing gender and social connectedness through community health improvement initiatives that lead to policy, systems, and environmental changes to improve gender norms and positively impact women’s health. Keippel et al. described the adoption of a city council of the Complete Streets resolution, which is informed by a gender lens. Healthy by Design further used gender information to successfully mobilise the community in response to threats of repeal of the policy and then influenced the adoption of a revised policy. This process resulted in a decrease of disparity between men and women in reported physical activity rates. On the other hand, Hardee et al. (2014) [32] found that structural interventions to address key social and structural drivers of HIV were successful in creating more gender-equitable relationships and decreasing violence; improving services for women; increasing widows’ ability to cope with HIV; and reducing behaviour that increases HIV risk. 

#### 3.1.3. Policy Evaluation

Thirteen papers carried out some type of evaluation of the results of the policies already implemented. These evaluations used quantitative methods (7), qualitative methods (2), both methods (2), and two were reviews. Of these, five evaluations were for health policies [18,25,26,33,35]. Of the ten studies on non-health policies, four focused on policies related to family and work [20,21,22,23]; four focused on policies to promote gender equity in the public sphere [23,25,30,32]; one focused on the gender identity law [19]; and one focused on any policy with implications for sexual and reproductive health [43] (Table 3). 

Most of the studies (9) [18,20,23,25,26,30,32,33,35] showed the persistence of gender inequalities in health. Three of them even indicated an increase of gender inequalities in access to health services and mortality in women [18,26,35]. Some analyses showed certain progress in gender equality (7) [19,21,22,23,25,32,43]. After the Gender Identity Law in Argentina was approved as a tool to reduce the marginalisation and exclusion that transgender people endure as a consequence of transphobia, positive changes were detected in particular domains such as education, health care, work, security, and civil rights [19]. Carael et al. found that a large majority of countries have integrated women-related issues into their national HIV policies and strategic plans (i.e., more countries than before reported that they have policies in place for ensuring that women and men have equal access to HIV-related services and reported having included women and girls as explicit target populations in their national strategic AIDS plans) [25]. Social and structural interventions on HIV had an impact on reduction of inequalities. These actions focused on different areas such as transformative gender norms; violence against women; transformative legal norms; promotion of women’s employment, income, and livelihood opportunities; and advance in education or reducing stigma and discrimination [32]. Regarding fertility, policies that ensure childcare coverage, family allowances, a higher prevalence of women’s part-time employment, and a higher length of paid leave were associated with completed fertility [21,22]. Nordic social democratic welfare regimes and dual-earner family models best promote women’s health, especially regarding the health of mothers [23]. Programs were designed in response to unequal gender norms and power dynamics, commonly aimed to increase women’s access to social and economic resources and/or participation in education and livelihoods programs, and improved women’s health status. 

This progress was mainly found in quality of life related to the gender identity law [19], the integration of gender equality in policies [32], improvement in diagnosis and treatment access for HIV [25], and health indicators related to fertility [21,22] and maternal health [23]. Moreover, Rottach et al. (2017) [43] identified safer and more equitable relationships, and marriage or birth of first child at a later age.

### 3.2. Recommendations

Eleven out of 33 studies included in our review made policy recommendations for the reduction of gender inequalities in health. Ten of them addressed recommendations regarding health policies [27,35,40,41,43,45,46,47,49,50] and six addressed recommendations regarding non-health policies [27,31,40,43,45,49]. With respect to methodology, seven used qualitative methods, one used mixed methods, and three were reviews. 

All of them advocated for the collection of gender and sex disaggregated data [27,31,35,40,41,43,45,46,47,49,50] in order to identify gender inequalities in health and to guide the design, implementation, and evaluation of the impact of policies to tackle them. This assignment could be entrusted to a gender observatory [41]. Seven papers emphasised the importance for decision-makers and other health professionals who implement policies to increase their gender awareness and capacity [27,35,39,40,41,46,50]. The authors stressed the need to analyse social factors in addition to biological ones [31,45], to address gender norms and structural factors [45,46,49], and to exchange best practices and lessons [27]. The authors underlined the need for sufficient financial and human resources [27,31,39,40,45,47,50] and research [27,31,35,43,45,47,49,50] to support gender-sensitive policies. In addition, they recommended going more deeply into the contextualisation of gender policies [31,47,49,50], as well as into their mechanisms and impact [35,43]. 

Four papers emphasised the need for alliances both between different health sectors and between professionals in other areas [27,43,46,50]. In addition, the importance of incorporating the active participation of women into the whole process was highlighted [27,31,40,41,49]. In the regulatory sphere, two papers defended the need to create a truly protective legal and political environment to reduce gender vulnerabilities [31,43] and to ensure compliance with commitments related to gender [35].

Finally, Forbes (2011) [27] developed very specific and detailed recommendations regarding the US national plan to combat HIV and other Sexually Transmitted Diseases (NHAS) to increase the gender sensitivity of the plan and to facilitate its implementation. Most of the proposals from the forum of professionals were specifications on already existing recommendations in the NHAS (Table 4).

## 4. Discussion

Our review brings together the available evidence on policies whose aim is to reduce gender inequalities in health. Despite growing evidence of worldwide gender inequalities in health and numerous recommendations to address them, our scoping review has shown that few policies have been formulated, implemented, or evaluated to tackle this problem. However, in spite of difficulties, some policies have achieved certain progress.

This scoping review has some limitations. As the aim was to be as sensitive as possible, the articles retrieved are highly heterogeneous, and this could hinder comparisons between studies. In addition, due to this heterogeneity, data extraction was complex. To maximise the quality of data extraction, it was carried out by two researchers, and any doubts or discrepancies were discussed and resolved jointly by the three researchers. Finally, the difficulty finding suitable articles led to problems establishing the final search strategy in order to balance sensitivity and specificity. In this attempt of balance, we may have missed a relevant publication. We tried to overcome this obstacle by including a hand search in the reference list of the final sample of included articles.

Our results showed that in recent years, publications on how to tackle gender inequalities have increased. However, the studies showed failures in design and particularly in the implementation process. Good intentions have not been implemented, and this is consistent with what has been criticised by the literature so far [15,51]. Some of the analysis pointed to essential gaps such as the lack of sex-disaggregated data and gender inequality indexes. This hinders any potential progress in the reduction of gender inequalities. To design effective policies, policy-makers need strategies to uncover gender-related health disparities in datasets with sex-disaggregated data. In order to address these gender inequalities in health, policy makers should go further than technical solutions and address the causes of these inequalities. This involves giving more space to social justice in policy making in order to tackle gender inequalities in health [52]. These long-standing issues are still neglected today as has been reported during the present SARS-CoV-2 pandemic [53]. However, although the WHO and other UN agencies with health-related mandates have built up banks of data and promising practices regarding what has worked or failed when addressing gender disparities in health [54], this valuable information remains in store without being used or analysed further. 

Twenty-five years ago, the Beijing Declaration and Platform for Action prompted work on gender equality. Subsequently, several declarations and agreements that have been made at the global level have incorporated the reduction of gender inequalities and women’s empowerment as an objective. The 2030 Agenda for Sustainable Development demands greater attention to the social determinants of health, including gender and multisectoral programming. However, the same barriers and constraints that were recognised by the Beijing signatories still exist, and the overall picture is of slow and unequal progress. This lack of development is related to under-financing, lack of capacity, bureaucratisation, shortage of relevant data, absence of women’s participation in decision-making, and difficulties in policy implementation and therefore, in achieving the desired reduction in gender inequalities. 

In spite of this lack of real commitment, there is some evidence of improvements that policy makers could consider implementing. Successful policies and recommendations based on previous experiences could guide future policies aimed at reducing gender inequalities in health. Our review did not aim to provide solutions but rather to gather evidence that supports the design of policies to reduce gender inequalities and recommendations that can facilitate their implementation. Policy makers and commissioners should be cautious. Scaling up these successful experiences will require an understanding of the sociocultural and political contexts in each country and each context. Greater efforts are needed to research, contextualise, and extend these policies and recommendations. There is a need for greater investment in implementation science to develop interdisciplinary methods and tools in order to develop and evaluate gender-sensitive health policies [55]. Mainstream public health and public policy have yet to invest substantially in research and action to tackle gender inequalities in health [56] and overcome the insufficient participation of women in the policy decision-making process.

## 5. Conclusions

There has been a great effort to include the reduction of gender inequalities on the public health agenda. Despite the normative efforts of defenders of women’s rights and equity gender, there have been insufficient resources, weak organisational mechanisms, and deficient political commitment, which have resulted in fragmented efforts and severe gaps between political rhetoric and political action.

Our scoping review has shown that the policies, aimed at reducing gender inequalities in health, either implemented or formulated as proposals, are scarce in the literature at the international level. Ten years ago, the United Nations stated that “we know what works” when taking care of women’s and children’s health. However, several barriers, including lack of gender awareness and policy makers’ lack of knowledge, continue to hinder progress. We need to learn from previous successful policy change experiences and to work on challenges if gender equity-oriented policies are to advance. Moreover, we need to include women’s voices in the process. There is an urgent need to act. 

## Figures and Tables

**Figure 1 ijerph-18-00327-f001:**
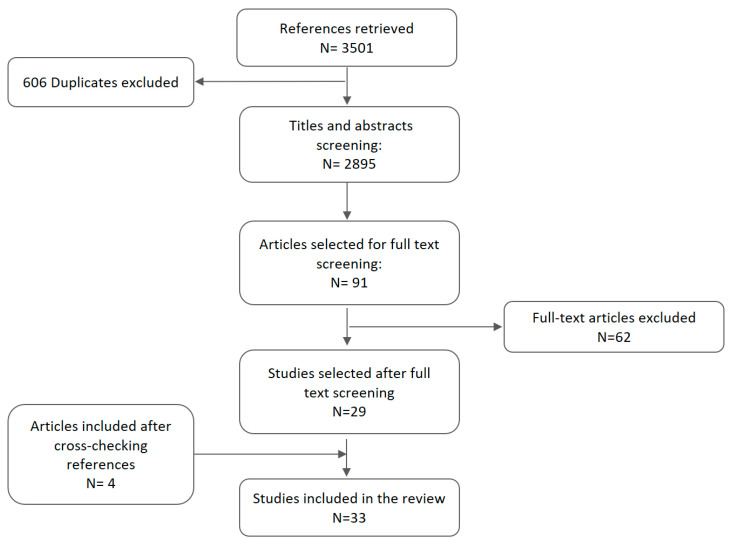
Flow chart of our study selection process.

**Table 1 ijerph-18-00327-t001:** Description of main characteristics and key outcomes of the 11 included papers analysing the formulation of policies aimed at reduce gender inequalities in health.

Policy Formulation (n = 11)			
		N	References
**Type of policy**			
	Health	7	[18,25,29,34,37,38,41]
	No health	5	[25,28,43,44,48]
**Methodology**			
	Quantitative	2	[25,38]
	Qualitative	8	[18,28,29,34,37,41,44,48]
	Review	1	[43]
**Key outcomes**			
	The non-inclusion of the importance of gender inequities in health in the justification of the policy	1	[38]
	Simplification of the causes of inequalities	2	[41,44]
	Lack of strategies to address inequalities	1	[43]
	Limitation of gender issues to maternal, child, and reproductive	3	[29,37,38]
	Lack of an approach to the legal rights of women	5	[18,25,29,43,44]
	Lack of participation of civil society and expert women’s associations	3	[29,41,44]

**Table 2 ijerph-18-00327-t002:** Description of main characteristics and key outcomes of the 16 included papers analysing the implementation of policies aimed at reducing gender inequalities in health.

Policy Implementation (n = 16)			
		N	References
**Type of policy**			
	Health	12	[18,25,29,33,35,37,38,39,40,41,42,50]
	No health	7	[19,24,25,32,36,39,40]
**Methodology**			
	Quantitative	2	[25,38]
	Qualitative	10	[18,19,29,36,37,39,40,41,42,50]
	Mixed	3	[24,33,35]
	Review	1	[32]
**Key outcomes**			
	Lack of will and political commitment	7	[18,29,35,39,40,42,50]
	Strong foothold of gender norms and laws that favour the lack of protection of women	4	[18,25,32,39]
	Insufficient awareness and capacity	8	[19,25,33,39,40,41,42,50]
	Strong vulnerability due to changes in organisations	4	[29,40,41,42]
	Not enough funding and human resources for implementation	5	[25,39,40,42,50]
	Bureaucratisation	3	[38,41,42]
	Lack of gender indicators for evaluation	5	[29,35,37,39,50]
	Lack of collaboration and absence of alliances	2	[37,50]
	Non-participation of civil society	4	[29,39,40,41]

**Table 3 ijerph-18-00327-t003:** Description of main characteristics and key outcomes of the 13 included papers evaluating policies aimed at reduce gender inequalities in health.

Evaluation of the Policy (n= 13)			
		N	References
**Type of policy**			
	Health	5	[18,25,26,33,35]
	No health	10	[18,19,20,21,22,23,25,30,32,43]
**Methodology**			
	Quantitative	7	[20,21,22,23,25,26,30]
	Qualitative	2	[18,19]
	Mixed	2	[33,35]
	Review	2	[32,43]
**Key outcomes**			
	Persistence of gender inequalities	9	[18,20,23,25,26,30,32,33,35]
	Reduction of gender inequalities	7	[19,21,22,23,25,32,43]

**Table 4 ijerph-18-00327-t004:** Policy recommendations to reduce gender based health inequalities identified in the articles analysed.

Recommendations (n= 11)			
		N	References
**Type of policy**			
	Health	10	[27,35,40,41,43,45,46,47,49,50]
	No health	6	[27,31,40,43,45,49]
**Methodology**			
	Qualitative	7	[27,31,40,41,47,49,50]
	Mixed	1	[35]
	Review	3	[43,45,46]
**Main Recommendations**			
	To collect gender and sex disaggregated data	11	[27,31,35,40,41,43,45,46,47,49,50]
	To conduct research to support gender-sensitive policies, including the mechanisms and impact of these policies	8	[27,31,35,43,45,47,49,50]
	To increase awareness and capacity among decision-makers and professionals	7	[27,35,39,40,41,46,50]
	To provide sufficient financial and human resources	7	[27,31,39,40,45,47,50]
	To increase participation of women in the development of gender sensitive policies	5	[27,31,40,41,49]
	To facilitate alliances and collaborations	4	[27,43,46,50]
	To contextualise gender policies	4	[31,47,49,50]
	To address gender norms and structural factors	3	[45,46,49]
	To create a truly protective legal and political environment for women	2	[31,43]
	To develop regulations to ensure political compliance	1	[35]
	To analyse both social and biological factors	2	[31,45]
	To exchange best practices and lessons	1	[27]
	To create a gender observatory	1	[41]

## Data Availability

All data is presented in this article. Researchers can contact authors regarding any request about the data.

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
