# Peer review of "Have Policies Tackled Gender Inequalities in Health? A Scoping Review"

_ijerph, 2021, doi:10.3390/ijerph18010327_

Round 1

Reviewer 1 Report

This is very well written. Especially the analytic strategy and data collection are very well organized and clear. That would be much more informative and clear if you could improve the following: 

In the Introduction, to strengthen the background of this research effort,  more specific ideas about general inequality should be described, such as the morbidity stand mortality rates by gender across countries, any aspects  of changes in the gender inequality of health in the past 10 years, 

Please briefly spell out Arksey and O'Malley's methodological framework. 

Author Response

Response to Reviewer 1 Comments

This is very well written. Especially the analytic strategy and data collection are very well organized and clear.

Thank you so much for your kind commentary.

That would be much more informative and clear if you could improve the following: 

- In the Introduction, to strengthen the background of this research effort, more specific ideas about general inequality should be described, such as the morbidity stand mortality rates by gender across countries, any aspects of changes in the gender inequality of health in the past 10 years.

In order to include more information to strengthen the background we modify 2 paragraphs of Introduction.

 Page 1, First paragraph of introduction, lines 31-36:

“Men die younger than women. In some countries that life expectancy gap is 11.7 years or more and is in part explained by their higher rates of consumption of tobacco and alcohol, their likelihood of death from violence, deaths from road injuries, and deaths by suicide. Even women might live longer, they suffer longer with chronic diseases. This higher risk of morbidities is linked to their reproductive roles, a lower status in society, and gender norms that impair their agency and bargaining position in sexual relationships”

Page 2, 45-46

“International commitments to advance gender equality have brought about improvements in some areas: child marriage and female genital mutilation have declined in recent years, and women’s representation in the political arena and in managerial positions is higher than ever before”

- Please briefly spell out Arksey and O'Malley's methodological framework. 

Following your suggestions we describe Arksey and O'Malley's framework in lines 87-91:

“Rather than being guided by a highly focused research question and search for a particular study design, the scoping study seeks to identify all relevant literature regardless of study design. In summary the main stages of a scoping review are: 1) to identify the research question; 2) to identify relevant studies; 3) study selection; 4) charting the data 5) collating, summarizing and reporting the results.”

Reviewer 2 Report

Important topic that addresses a gap in the literature. Generally well written.

The characteristics of the 29 included papers extracted should be added as a supplementary table. As it is Table 1- 4 shows the details in a disjointed way.

Mention should be made on what the policies were and not just the outcomes.

What were the differences between the different countries/ regions assessed in the reviews. Without looking at each reference article included- its hard to establish this. what were the policies that showed some success?

Author Response

Response to Reviewer 2 Comments

Important topic that addresses a gap in the literature. Generally well written.

Thank you so much for your kind commentary.

The characteristics of the 29 included papers extracted should be added as a supplementary table. As it is Table 1- 4 shows the details in a disjointed way.

Mention should be made on what the policies were and not just the outcomes. What were the differences between the different countries/ regions assessed in the reviews. Without looking at each reference article included- its hard to establish this. what were the policies that showed some success?

Considering your suggestion we included 2 supplementary files which include the main characteristics of the reviewed articles. The supplementary file 2 includes the studies that analysed policies and the supplementary file 3 includes the studies that made recommendations. In addition, the supplementary file 2 contains a description of the policies analysed. Finally, in Results section, we also included a brief description of the policies with certain success.

Lines 184-191: “Burke et al analysed the community coalition created to develop and incorporate strategies for addressing gender and social connectedness through community health improvement initiatives that lead to policy, systems, and environmental changes to improve gender norms and positively impact women’s health.  Keippel et al described the adoption of a city council of the Complete Streets resolution, informed by a gender lens. Healthy by Design further used gender information to successfully mobilize the community in response to threats of repeal of the policy, and then influenced the adoption of a revised policy. This process resulted in a decrease of disparity between men and women in reported physical activity rates has decreased”

Lines 210-227: “After the Gender Identity Law in Argentina was approved as a tool to reduce the marginalization and exclusion that transgender people endure as a consequence of transphobia, positive changes were detected in particular domains such as education, health care, work, security, and civil rights [17]. Carael et al found that a large majority of countries have integrated women-related issues into their national HIV policies and strategic plans (i.e. more countries than before reported that they have policies in place for ensuring that women and men have equal access to HIV-related services and reported having included women and girls as explicit target populations in their national strategic AIDS plans) [23]. Social and structural interventions on HIV had an impact on reduction of inequalities. These actions focused on different areas like transformative gender norms; violence against women; transformative legal norms; promotion of women’s employment, income and livelihood opportunities; advance in education or reducing stigma and discrimination [30].  Regarding fertility, policies that ensure childcare coverage, family allowances, a higher prevalence of women’s part-time employment and higher length of paid leaves were associated with completed fertility [19,20]. Nordic social democratic welfare regimes and dual-earner family models best promote women’s health, especially regarding the health of mothers [21]. Programes designed in response to unequal gender norms and power dynamics, commonly aimed to increase women’s access to social and economic resources and/or participation in education and livelihoods programs, and improved women health status.”

We must say due to heterogeneity of the results is hard to establish differences or patterns between countries.

Reviewer 3 Report

The manuscript is offering some original ideas in the field of health research. This is a very well written and well searched scoping review.

The authors could have indicated in more detailed why this scoping exercise is needed and what are the expected outcomes of the research.

They should also carefully check whether they missed any important relevant research papers.

Author Response

Dear reviewer,

Thank you so much for your review. We detail the response to your comments in the following attached document. Please see the attachment.

Best regards,

Nuria Crespí-Lloréns
